# Topological entanglement properties of disconnected partitions in the Su-Schrieffer-Heeger model

**Tommaso Micallo**[1,2,3,4*], **Vittorio Vitale**[2,4],
**Marcello Dalmonte**[2,4] **and Pierre Fromholz**[2,4†]

**1** Institute of Theoretical Physics, Technische Universität Dresden, 01062 Dresden, Germany
**2** The Abdus Salam International Centre for Theoretical Physics,
strada Costiera 11, 34151 Trieste, Italy
**3** Dipartimento di Fisica, Università di Trento, 38123 Povo, Italy
**4** International School for Advanced Studies (SISSA), via Bonomea 265, 34136 Trieste, Italy

⋆ tommaso.micallo@tu-dresden.de, † fromholz@ictp.it

## Abstract

We study the disconnected entanglement entropy, $S^D$, of the Su-Schrieffer-Heeger model. $S^D$ is a combination of both connected and disconnected bipartite entanglement entropies that removes all area and volume law contributions and is thus only sensitive to the non-local entanglement stored within the ground state manifold. Using analytical and numerical computations, we show that $S^D$ behaves like a topological invariant, *i.e.*, it is quantized to either $0$ or $2\log(2)$ in the topologically trivial and non-trivial phases, respectively. These results also hold in the presence of symmetry-preserving disorder. At the second-order phase transition separating the two phases, $S^D$ displays a finite-size scaling behavior akin to those of conventional order parameters, that allows us to compute entanglement critical exponents. To corroborate the topological origin of the quantized values of $S^D$, we show how the latter remain quantized after applying unitary time evolution in the form of a quantum quench, a characteristic feature of topological invariants associated with particle-hole symmetry.

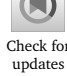

# 1 Introduction

When a phase is topological, then its ground state(s) displays robust entanglement properties. The converse is more uncertain: to what extent are entanglement properties unique to topological states? This attempt at understanding topological phases through the lens of entanglement is recent [1]. It has been successful for (true) topological order that is now characterized by the topological entanglement entropy (TEE) [2–4]. This quantity works both in- [5–8] and out-of-equilibrium [9–11], and is included in the textbooks' definition of these phases [12,13]. It provides a useful discriminating characterization of topology for numerical simulations [14–16] and it stimulated the search for corresponding experimental entanglement probes [17–23].

Topological insulators and superconductors or, more generally, symmetry-protected topological phases (SPTP) also display characteristic entanglement features. Amongst these features, the most used is the entanglement spectrum [24–26]. It serves as an entanglement-based *sine qua non* signature of an SPTP. This spectrum corresponds to all the eigenvalues of the bipartite reduced ground state's density matrix of the system. The degeneracy of the matrix' few largest eigenvalues is imposed by the dimension of the possible representations of the edge states [24,25,27,28]. Because the same spectrum may come from a non-topological state, the diagnosis it provides is a necessary but not sufficient condition [1].

The disconnected entanglement entropy $S^D$ is another entanglement signature for SPTPs of systems with open boundary conditions. $S^D$ was suggested and tested through simulations for some examples of bosonic topological phases in Ref. [29]. Like the TEE, it extracts a topological-exclusive contribution to the bipartite entanglement entropy. Unlike the TEE [4, 30], this contribution is not (yet) predicted by quantum field theory as it is related to short-range or edge-edge entanglement. Ref. [31] used the Kitaev wire [32] to prove that $S^D$ is also valid for 1D topological superconducting phases, where it is captured within a lattice gauge theory framework. In the Kitaev wire, $S^D$ is traced back to the entanglement necessarily present by construction between the only two fractional (Majorana) modes of the model, even when the modes are localized on each edge of the chain.

---

[1]An example is the entanglement spectrum of the topological ground state of the Kitaev wire, which is identical to the spectrum of the corresponding ground state of the non-topological ferromagnetic spin 1/2 Ising chain after the Jordan-Wigner mapping.

This paper aims at characterizing the properties of $S^{\mathrm{D}}$ for the case of one-dimensional topological insulators, focusing on a simple, yet paradigmatic example: the Su-Schrieffer-Heeger model [33, 34] (SSH). This model of spinless fermions displays a topologically trivial phase and an SPTP with two edge modes. Each of these states is usually represented as one dangling fermion unentangled with the bulk on either side of the chain. The bulk is short-range entangled [35]. However, the finite size of the chain ensures a systematic maximal entanglement between the two a priori independent edge states as predicted in bosonic topological phases [36, 37]. As we show below, $S^{\mathrm{D}}$ is sensitive to this long-range entanglement and takes the maximal possible value of $2\log 2$ in the SPTP [2]. In contrast, this value is 0 in the trivial phase. Therefore, our first claim is that $S^{\mathrm{D}}$ can be a good signature of topology for SPTP without fractional edge states like in the SSH model.

We also find that $S^{\mathrm{D}}$ provides additional quantitative topological exclusive information on the entanglement properties of the ground state. Indeed, $S^{\mathrm{D}}$ displays a system-size scaling behavior close to the critical phase transition, akin to the magnetization of an Ising chain. We obtain the resulting critical exponents using exact numerical methods. Within the topological phase, $S^{\mathrm{D}}$ remains quantized on average in the presence of disorder. Such a scaling behavior and critical exponents are different with respect to the Kitaev wire [31] (the only other occurrence of such an analysis for fermionic systems to our knowledge) and to bosonic cluster models realized as instances of random unitary circuits (e.g., Ref. [39]). This indicates that while scaling behavior is likely a generic feature of $S^{\mathrm{D}}$ at criticality, the corresponding critical entanglement exponents depend on the nature of the associated topological phase transition.

Finally, we apply unitary evolution to the system in the form of quantum quenches either within the two phases or across the phase transition. After the quench, we observe that $S^{\mathrm{D}}$ keeps its initial value in the limit of an infinite chain, a behavior characteristic of a topological invariant associated with particle-hole symmetry [40]. This set of observations mimics the phenomenology observed for the TEE in the context of true topological order and allows us to define $S^{\mathrm{D}}$ as a valid entanglement order parameter.

This work is structured as follows. We introduce the SSH model and the disconnected entanglement entropy in Sec. 2. In Sec. 3, we present the analytical computation of $S^{\mathrm{D}}$ in the topological phase, trace it back to the systematic maximal entanglement of the edge states, and explain its exponential finite-size scaling. In Sec. 4, we present our numerical results for the case of the ground state of a clean SSH chain. In Sec. 5, we investigate quantum quench protocols, that provide a clear characterization of $S^{\mathrm{D}}$ as a topological invariant. In Sec. 6, we showcase one application of $S^{\mathrm{D}}$, by investigating the entanglement properties of disordered SSH chains, and showing how the disconnected entropy recovers the predicted phase diagram. We discuss the generality of our findings within the BDI and D classes of the tenfold-ways in Sec. 7, and conclude the study in Sec. 8.

## 2 Model Hamiltonian and disconnected entropies

We briefly introduce the SSH model and both its topological and trivial phases in Sec. 2.1. We introduce $S^{\mathrm{D}}$ in Sec. 2.2. In Sec. 2.3, we confront the strengths and the limits of $S^{\mathrm{D}}$ that become apparent for the SSH model with periodic boundary conditions. We establish the equivalence of using $S^{\mathrm{D}}$ with either the von Neumann and Rényi-2 entanglement entropies for the SSH model in Sec. 2.4.

---

[2]This result is consistent with a previous study of the spinfull interacting SSH model in Ref. [38]. Another quantity is used then that also extracts the edge entanglement and coincides with $S^{\mathrm{D}}$ for this model.

## 2.1 The SSH model

The Su-Schrieffer-Heeger model [33,34] describes a one-dimensional spinless fermionic chain with a staggered hopping between sites. The chain is composed of $N$ unit cells. Each cell is divided into one site $A$ connected to one site $B$. The number of sites in a chain is, hence, $L = 2N$. The Hamiltonian of the model with open boundary conditions is:

$$H_{\text{SSH}} = -v \sum_{i=1}^{N} \left( c_{iA}^{\dagger} c_{iB} + h.c. \right) - w \sum_{i=1}^{N-1} \left( c_{i+1A}^{\dagger} c_{iB} + h.c. \right), \tag{1}$$

where $c_{iX}^{\dagger}$ ($c_{iX}$) is the creation (annihilation) operator of a spinless fermion on the unit cell $i$, site $X = A, B$. $v > 0$ ($w > 0$) is the intra-(inter-)cell hopping amplitude. The chain is represented in Fig. 1a).

At half-filling, the model displays two phases. When $v/w \gg 1$, $A$ and $B$ within a unit cell dimerize and the phase is topologically trivial. There is one particle per unit cell (box in Fig. 1a)). The vanishing entanglement between the unit cells increases until $v/w = 1$ where the phase transition occurs. When $v/w < 1$, the phase is topological. The single-particle spectrum displays two zero-mode edge states in the gap between two bands. When $v/w \ll 1$, the density profile of each edge state shows one localized fermion in the leftmost or rightmost site of the chain. The dynamics of this fermion is independent of symmetry-preserving perturbations of the bulk. Since Pauli's filling rule applies, at half-filling and zero temperature, all the lower band is occupied. The ground state is unique and its bulk is short-range entangled: one fermion forms a Bell pair on each strong link (darkest line in Fig. 1a) ). When $v = 0$, the Bell pair on the link between cell $i$ and $i+1$ is strictly localized. The Bell pair can be expressed as:

$$\left( |0_{iB} 1_{i+1A}\rangle + |1_{iB} 0_{i+1A}\rangle \right) / \sqrt{2}. \tag{2}$$

For a finite chain with open boundary conditions, the ground state has one fermion populating each strong link, and one extra fermion in the superposition of Eq. 2 of the two edge states (see Sec. 3). This superposition entangles the two distant edges maximally. It is the only long-range entanglement in the system and contributes to the entanglement entropy. We will show that this is the contribution extracted by $S^{\text{D}}$ (Sec. 2.2). The topological and the trivial phases are separated by a critical phase transition at $v/w = 1$.

The topological invariant of this model is the Zak phase [41], a quantity proportional to the Berry phase [42]. By definition, a topological invariant is constant and quantized over the whole phase and only changes across a phase transition. Here, the Zak phase distinguishes the two regimes of $v/w$ while the way their ground states at half-filling breaks the initial symmetries can not. This situation is beyond the spontaneous symmetry breaking paradigm and signals that at least one of the two phases is topological. For open boundary conditions, the topological regime is the only one displaying edge states (and a non-zero Zak phase). For periodic boundary conditions, the values of the Zak phase for the two regimes can be exchanged with the *renaming*:

$$A \to B, \quad v \to w, \tag{3a}$$

$$B \to A, \quad w \to v, \tag{3b}$$

with unchanged values of the amplitudes. In this case, the Zak phase only ensures that the two phases are topologically distinct.

The topological edge states are protected by charge conservation (U(1) symmetry), the time-reversal $T$, the particle-hole $C$, and the chiral $S$ (or rather, the sublattice) symmetry [3].

---

[3]There are several definitions of these symmetries, leading to several self-consistent 10-fold ways. We use the definitions of Ref. [43].

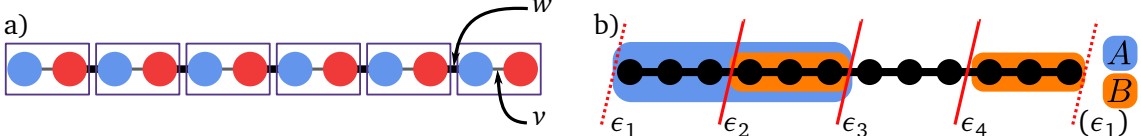

Figure 1: (Color online) The SSH chain and the partition used for $S^D$. a) The SSH chain is a spinless fermionic chain of sites $A$ and $B$ (blue and red on the figure) connected with staggered hoppings $v$ and $w$. A pair of sites $A$ and $B$ forms a unit cell (a box on the figure). b) Partitions A (shaded, blue) and B (shaded, orange) associated with the entanglement topological order parameter $S^D$. For large systems, the exact locations of the cuts $\epsilon_i$, $i = 1, 2, 3, 4$, of each partition do change the value of the bipartite entanglement entropy but do not change $S^D$.

The phase is part of of the BDI-class of the Altland-Zimbauer ten-fold ways [44] leading to a $\mathbb{Z}$ classification. This classification indicates that there are an infinite countable number of distinct topological phases with unbroken $T, C$ and $S$ in 1D. The classification will be $\mathbb{Z}_8$ if symmetry-preserving interactions are allowed [45]. This latter classification means that there are only seven non-trivial topological phases left. While the translation symmetry is needed to compute some non-local topological order parameters such as the winding number in $k$ space, the topological phase is not protected by this last symmetry.

In simulations, the Zak phase, the winding number, the presence of edge states, or the entanglement spectrum have all served as smoking guns for topology. In experiments, the winding number was measured [46, 47] for the SSH model and its generalization [48]. In these instances, the winding number is contained in the time evolution of the chiral mean displacement observable. This observable quantifies the relative shift between the two projections of the tracked state onto the eigenstates of the chiral operator [47]. It follows a random walker [47] or the entire atomic population after a sudden quench [46, 48]. The observable is measured using chiral- and site-resolved imaging over several times. $S^D$ completes this list of topological detectors. Amongst them, $S^D$ is the only probe that is both unambiguous and entanglement-based [4]. Thus, $S^D$ is the optimal tool to study the topological entanglement properties of the SSH model.

## 2.2 The *disconnected* entanglement entropy $S^D$

The *disconnected* entanglement entropy $S^D$ has been introduced in Ref. [29] as a generalization of the topological entanglement entropy $S_{\text{topo}}$ for all the topological phases (topological order and SPTP). $S_{\text{topo}}$ is an exclusive marker for topological orders. Both $S^D$ and $S_{\text{topo}}$ aim to isolate a constant topological-exclusive contribution in the bipartite entanglement entropy. Thus, they are both built using the same linear combination of entropies but they differ in the partitioning of the system. In both cases, the combinations are chosen to cancel volume law (linear in system size) and area law (linear in the number of internal cuts) contributions in the bipartite entanglement entropy. For $S_{\text{topo}}$, the leftover constant contribution is topological-exclusive, as shown by topological quantum field theory arguments [4, 30]. A similar generic proof is missing for $S^D$. Instead, simulations, exact solutions, or a gauge theory analogy validate its success for some examples [29, 31, 50].

The original definition of $S^D$ [29] uses the von Neumann entanglement entropy of a bipartition of the system. For a chain divided into two complementary subsets $A$ and $\bar{A}$, the reduced density matrix of the subset $A$ is

$$\rho_A = \text{Tr}_{\bar{A}} \rho.$$

---

[4]Another entanglement-based topological detecting quantity that applies to the SSH model would be the boundary susceptibility [49].

$\text{Tr}_{\bar{A}}$ stands for the partial trace on the subset $\bar{A}$, and $\rho$ is the full (pure) density matrix of the system, always taken as a ground state in this study. $A$ may be a collection of disconnected sites of the chain. The von Neumann bipartite entanglement entropy follows:

$$S_A = -\text{Tr}_A \rho_A \log(\rho_A). \tag{4}$$

$S^D$ uses the partitioning of the chain in Fig 1b), so that:

$$S^D = S_A + S_B - S_{A\cup B} - S_{A\cap B}. \tag{5}$$

The lengths of $A$, $B$, and the disconnected subset $D = \overline{A \cup B}$ are respectively $L_A$, $L_B$, and $L_D$.

The formula is best understood when comparing what happens for the other possible gapped phases in 1D: disordered (paramagnetic) and ordered phases, the latter ones characterized by some of form of spontaneous symmetry breaking (SSB) of a discrete symmetry.

A trivial phase always has one single ground state independently of its boundary conditions and for both the thermodynamical limit (large number of particle) and finite (large) size. When this state is a product state, any choice of partition leads to a bipartite entanglement entropy of zero. An example of such a case is the ground state of a (quantum) spin-1/2 chain with only a magnetic field. The ground state can also be short-range entangled: if $C_1$ and $C_2$ are two simply connected partition of the chain separated by a large distance, then

$$\rho_{C_1 C_2} \sim \rho_{C_1} \otimes \rho_{C_2}. \tag{6}$$

The trivial SSH phase is an example of this scenario. In that phase, the mutual information $I(C_1 : C_2)$ of two disjoint and distant partitions is zero, and so is $S^D$ for open boundary conditions (the *conditional* mutual information in this context). Indeed, for large subsets in Fig. 1b), Eq. 6 applies, such that :

$$S^D = S_A + \left(S_{A\cap B} + S_{B\setminus A}\right) - \left(S_A + S_{B\setminus A}\right) - S_{A\cap B}, \tag{7a}$$
$$= 0, \tag{7b}$$

for $A$ and $B$ the partitions in Fig. 1b), and where $B\setminus A$ means $B$ without $A \cap B$. For periodic boundary conditions, both $A$ and $B\setminus A$ are connected such that $S^D = I(A : (B\setminus A))$. For these conditions, $S^D$ may vary within a phase.

A SSB phase always has several 'ground states' independently of its boundary conditions. A basis of these states may be expressed as product states. For a finite-size systems, the degeneracy is lifted by corrections that are exponentially small in system size: the single ground state has a GHZ-type of quantum entanglement [29]. An example of a SSB phase is the Ising chain with a small transverse magnetic field. The true finite-size ground state is then the maximally entangled symmetric superposition between the state with all spin up and all spin down: the GHZ-state. Any bipartite entanglement entropy of this state has the same non-zero value such that all the entropies in Eq. 5 are equal and $S^D = 0$ [5]. Like in the trivial case, additional short-range entanglement in the ground state does not change $S^D$.

For periodic boundary conditions, the ground state of an SPTP is unique. This state is short-range entangled after defining the proper unit-cell. Like the periodic trivial case, $S^D = I(A : (B\setminus A))$ then. Unlike the trivial case, the SPTP imposes neighboring unit cells to be maximally entangled, saturating $S^D$ (cf Sec. 2.3) so that it will not vary within the same phase. For open boundary conditions, an SPTP displays edge (zero)-modes. A basis of these modes can sometimes be written as separable states, like the SSB case [6]. Unlike the SSB case,

---

[5]When $L_D = 0$, $A \cup B$ spans over the whole system. In this case, the combination Eq. 5 reduces to the *tripartite* entanglement entropy [29], and is non-zero for both SSB and SPT phases.

[6]If we define separable in terms of sites, the edge states are separable for the topological SSH, but they are not separable for the topological Kitaev wire. In terms of Majorana fermions, both are separable. In terms of unit cells, neither are separable. Interactions typically prevents separability.

the edge modes all have the same bulk, and the superposition of the same bulk does not increase the entanglement. For a SSH chain with two edges, the true ground state is a maximally entangled superposition of these edge states. This superposition generates an additional saturated contribution (i.e., of maximal possible value) to the entanglement entropy of a partition that includes one edge without the second (like $A$ and $B$ in Fig. 1b)). $S^D$ then behaves like in Eq. 7a, but with this extra edge contribution for $S_A$ and $S_B$ that is not compensated by $S_{A \cup B}$ and $S_{A \cap B}$.

Only this edge contribution sets $S^D$ to a quantized, non zero value. Similarly to Ref. [38], the value of $S^D$ in the thermodynamical limit is fixed by the number of edge states $\mathcal{D}$ (or, equivalently, by the dimension $2^{\mathcal{D}}$ of the Hilbert space they span):

$$\lim_{L \to \infty} S^D = 2 \log \mathcal{D}. \tag{8}$$

$\mathcal{D}$ is fixed by the bulk-edge correspondence outside of the accidental increase of global symmetry due to fine-tuning. Thus, $\mathcal{D}$ is almost a robust topological invariant, and so is $S^D$. The SSH topological phase fits in this scenario.

Thus, the SPTP case can be interpreted as a trivial gapped phase with saturated short-range entanglement in the bulk, with an extra entanglement between the edge states for a finite open system.

## 2.3 Periodic boundary conditions

Following the previous discussion, we will solely focus on open boundary conditions in the rest of the text. We take a brief detour in this section to discuss the physical interpretation of $S^D$ for close chains.

For periodic boundary conditions and at half-filling, $S^D$ only extracts the saturated entanglement of the cut between the connected partitions $A$ and $B \backslash A$ of the single bulk ground-state. The saturation comes from cutting the singlet between two neighboring projective representations on each side of the cut. This picture stems from the cohomology and supercohomology classification [51, 52]. It means that if $G$ is the unbroken symmetry group of the chain, then cutting a chain between two unit cells leaves an edge state on each side of the cut. One edge state transforms according to a projective representation of $G$, and the other transforms according to the conjugate representation of the former edge. When connected back, the two edge states form a singlet that is maximally entangled by construction. This topological pattern repeats all along the chain and explains the saturation of the bulk short-range entanglement.

The projective representations involved in this internal cut also transforms the edge states of the chain with open boundary conditions. $S^D$ has thus the same saturated value for both boundary conditions in the topological phase. In contrast, $S^D$ is systematically zero only for the trivial phase of a system with open boundaries. Therefore a sharp phase transition between the two phases only exists for open boundary conditions.

This structure is explicit in the SSH model: the two phases for $v/w < 1$ and $v/w > 1$ are a collection of coupled dimers between $B$ and $A$ or $A$ and $B$ (the order matters) respectively. These dimers become uncoupled when $v = 0$ and $w = 0$ respectively. The contribution to the entanglement entropy for any bipartition of the system then corresponds to the contribution of each cut: $\log 2$ for a cut in the middle of a dimer and 0 otherwise. Defining $\epsilon_j = 1$ when the cut $j = 1, 2, 3, 4$ (see Fig. 1b)) separates a dimer and $\epsilon_j = 0$ when the cut is between two of them, Eq. 5 becomes:

$$\begin{aligned} S^D / \log 2 &= \left( (\epsilon_1 + \epsilon_3) + (\epsilon_1 + \epsilon_2 + \epsilon_3 + \epsilon_4) - (\epsilon_3 + \epsilon_4) - (\epsilon_2 + \epsilon_3) \right), \\ &= 2 \epsilon_1. \end{aligned} \tag{9}$$

Eq. 9 is not one-site translation-invariant for the two phases. This lack of invariance stems from the ambiguity highlighted in Eq. 3 and is only lifted after defining the unit cell and always cutting between two of them. Note that $\epsilon_1$ is exactly the quantity extracted by the "edge entanglement entropy" of Ref. [38] when there are no volume nor GHZ-like contributions in the bipartite entanglement entropy. $\epsilon_1$ is also the contribution in the value of the bipartite entanglement entropy that is linked to the Zak phase in the small localization length and thermodynamical limit [53]. Consequently both $\epsilon_1$ and the Zak phase change depending on the definition of the unit cell.

## 2.4 Disconnected Rényi-2 entropy

Similarly to the bipartite entanglement entropy, it is possible to define and use the disconnected entropy using the Rényi-$\alpha$ entanglement entropies [31]. These extensions are useful for two reasons. First, for small values of $\alpha$, the Rényi-$\alpha$ entanglement entropies are experimentally measurable. Second, for $\alpha = 2$ (and, with increasing complexity, for larger integer values of $\alpha$ as well), they can be computed using Monte Carlo methods, providing a natural framework to extend our methods to interacting systems.

The Rényi-$\alpha$ entanglement entropy [54] of a bipartition $A, \bar{A}$ of the chain is defined as:

$$S_{A,\alpha} = \frac{1}{1-\alpha} \log \operatorname{Tr}_A\left(\rho_A^{\alpha}\right), \tag{10}$$

where the case $\alpha \to 1$ is the von Neumann entanglement entropy. The subsequent versions of $S^{\mathrm{D}}$ are:

$$S_{\alpha}^{\mathrm{D}} = S_{A,\alpha} + S_{B,\alpha} - S_{A\cup B,\alpha} - S_{A\cap B,\alpha}, \tag{11}$$

for the partition of the chain in Fig. 1b). To motivate and then support the relation between $S^{\mathrm{D}}$ and $S_{\alpha}^{\mathrm{D}}$, we will make use of the following known properties:

1. For all $\alpha \in ]0, +\infty[$, $S_{\alpha}$ has the property of minimum value [55] (i.e., $S_{\alpha}(\rho) = 0 \iff \rho$ is a pure state). Hence every individual bipartition in Eq. 5 and Eq. 11 are simultaneously zero or non-zero, i.e., $S_{X,\alpha} \neq 0 \iff S_X \neq 0$.

2. For all $\alpha > 1$, $S_{\alpha}$ has the property of monotonicity [56], i.e., for $1 < \alpha_1 \leq \alpha_2$, $S_{\alpha_1}(\rho) \geq S_{\alpha_2}(\rho)$. This property can be extended to the von Neumann case $\alpha = 1$. All bipartite von Neumann entanglement entropy of 1D gapped isolated systems are finite, and thus, by monotonicity, so will be their Rényi-$\alpha > 1$ counterpart. Thus, there are no divergent terms in Eq. 11 for the SSH model with $w \neq v$ [7].

3. Despite open boundary conditions and for large enough subsets, translation invariance imposes equality of finite entropies of simply connected subset, i.e., $S_{A,\alpha} = S_{A\setminus B,\alpha} = S_{B\setminus A,\alpha}$ and $S_{A\cap B,\alpha} = S_{D,\alpha}$ (the presence of exactly one edge matters). With additional homogeneous disorder, the equalities become $\langle S_{A,\alpha} \rangle = \langle S_{A\setminus B,\alpha} \rangle = \langle S_{B\setminus A,\alpha} \rangle$ and $\langle S_{A\cap B,\alpha} \rangle = \langle S_{D,\alpha} \rangle$. A corollary follows: if $X$ and $Y$ are simply connected large subsets that include the same number of edges, then $S_X = S_Y \iff S_{X,\alpha} = S_{Y,\alpha}$. When the system is translation-invariant every two sites (or more) instead, like the SSH model, the value of a connected entropy changes depending on the position of its two cuts relatively to the unit cells. These changes are compensated in $S^{\mathrm{D}}$, similarly to how internal cuts compensate each others in Eq. 9. This difficulty can be bypassed by considering only the unit cells instead of the sites, and only the cuts between unit cells.

---

[7] For $\alpha \in ]0, 1[$, we must assume that each term in Eq. 11 will also be finite for all bipartition of 1D gapped system. Then, the rest of the demonstration applies, and $S_{\alpha}^{\mathrm{D}}$ can also be used for $\alpha \in ]0, 1[$. The Rényi-1/2 entanglement entropy is useful as it coincides with the logarithmic negativity for pure states.

4. For all $\alpha > 1$, $S_\alpha$ has the property of additivity. This property imposes $S_{B,\alpha} = S_{A \cap B,\alpha} + S_{B \backslash A,\alpha}$ for short-range entangled 1D systems.

The first point establishes the qualitative correspondence between the von Neumann and the Rényi-$\alpha$ bipartite entanglement entropies: one of the two entropies is zero if and only if the second is also zero. The von Neumann entanglement entropy never diverges for gapped phases. As a consequence, the second point prevents the Rényi entropies to diverge as well. The third point ensures that the considerations of Sec. 2.2 for the archetypal trivial phase and SSB phase stay valid for $S_\alpha^{\mathrm{D}}$. The fourth point extends this validity for the short-range entangled variations around the archetypal cases and the SPTP. The value of $S^{\mathrm{D}}$ and $S_\alpha^{\mathrm{D}}$ may differ by a finite factor $\gamma_\alpha$: $S^{\mathrm{D}} = \gamma_\alpha S_\alpha^{\mathrm{D}}$.

The von Neumann entanglement entropy is important in quantum information as it counts the maximum amount of distillable entangled pair between a subset and its complementary. Instead, the Rényi-2 entropy can be measured experimentally [19, 20, 57, 58] for all systems in all dimensions, so that $S_2^{\mathrm{D}}$ is experimentally measurable for small subset sizes. The disconnected part can be large. The sizes of $L_A = 8$ or 12 are both accessible experimentally [58] and enough to reach the saturated value of $S_2^{\mathrm{D}}$ in the simulations Sec. 4 to 6. In practice, measuring $S_2^{\mathrm{D}}$ is done by measuring each bipartite entanglement entropy in Eq. 5 successively (using the same system if needed).

# 3 Analytical predictions on $S^{\mathrm{D}}$: long-range entanglement between edges

In this section, we present an explicit calculation of $S^{\mathrm{D}}$ for the SSH model to justify the cartoon pictures of Sec. 2. In the topological phase, we show that the ground state always contains the maximally entangled superposition of the two localized edge states. This superposition ensures that $S^{\mathrm{D}} = 2 \log 2$ up to exponential corrections in the size of the system. $S^{\mathrm{D}} = 0$ for the non-topological phase. This result is valid only when the chain has two edges, i.e., for a finite chain of arbitrary large length.

We first show that the two edge states in the one-body spectrum are in the symmetric and antisymmetric superposition when the chain is finite but of arbitrary large length. We obtain the exact expressions of the two states for weak link hopping $v = 0$ (see Fig. 1a)) and track their change when $v$ increases [59]. This hopping $v$ slightly spreads the localized edge states and lifts the degeneracy between the two such that the symmetric superposition of the two is lower in energy. We thus observe the exponential convergence of $S^{\mathrm{D}}$ in $L$ to $2 \log 2$. This result is quantitatively consistent with the simulations (see Fig. 2b)) away from the phase transition ($v = w = 1$) and for large system size. Ref. [60] provides an exact but more involved derivation of the spectrum and eigenstates of the SSH chain for all $v$ and $w$.

Ref. [59] provides the detailed derivation followed in this section. We remind here the main steps and results. The SSH Hamiltonian with disorder is:

$$H_{\mathrm{SSH}}^{\mathrm{dis}} = -\sum_{i=1}^{N} v_i \left( c_{iA}^\dagger c_{iB} + h.c. \right) - \sum_{i=1}^{N-1} w_i \left( c_{i+1A}^\dagger c_{iB} + h.c. \right). \tag{12}$$

Since the Hamiltonian is non-interacting, a complete solution only requires the one-body spectrum and the corresponding eigenstates. At zero temperature, each state is filled in order of increasing energy until the target filling fraction is reached. Because of chiral symmetry, the spectrum is symmetric around $E = 0$, and when they exist, the edge states are the only states at that energy. It follows that the ground state of the topological phase corresponds to the full

lower band filled (i.e., all bulk dimers filled with one particle each), and one edge state populated. To express the wave function of latter, we consider the most generic one-body wave function:

$$|\Psi\rangle = \sum_{i=1}^{N}\left(a_i c_{i,A}^{\dagger} + b_i c_{i,B}^{\dagger}\right)|0\rangle, \tag{13}$$

where $|0\rangle$ is the particle vacuum and $a_i$ and $b_i$ are complex weights. This state is a zero-mode of the Hamiltonian Eq. 12 for the infinite chain. For the finite chain (of arbitrarily large length), the approximation $H_{SSH}^{dis}|\Psi\rangle = 0$ imposes (for $v_i, w_i > 0$):

$$\text{For } i = 2, ..., N, \qquad\qquad a_i = a_1 \Pi_{j=1}^{i-1}\frac{-v_j}{w_j}, \tag{14a}$$

$$\text{For } i = 1, ..., N-1, \qquad\qquad b_i = b_N \frac{-v_L}{w_i}\Pi_{j=i+1}^{N-1}\frac{-v_j}{w_j}, \tag{14b}$$

$$b_1 = a_N = 0. \tag{14c}$$

In the limit $N \to \infty$, Eqs. 14a and 14b reveals two states, $|L\rangle$ and $|R\rangle$. The two states are exponentially localized on either the first site $A$ or the last site $B$ of the chain with (average) localization length:

$$\xi = \frac{N-1}{\log\left(\Pi_{i=1}^{N-1}|w_i|/|v_i|\right)}. \tag{15}$$

The condition Eq. 14c is instead incompatible with the existence of zero-energy modes and one must consider the (small) lift in the degeneracy between the two edge states. In this case, the best approximations of the two edge states are the two orthogonal real equal-weighted superpositions of $|L\rangle$ and $|R\rangle$. When the cuts in Fig. 1b) are far apart both from each other and the boundaries, this superposition ensures $S^D = 2\log 2$ approached exponentially.

When one $v_i = 0$, the exponential tail of both localized edge state $|L\rangle$ and $|R\rangle$ is truncated at site $i$. When instead one $w_i = 0$, the exponential tails also stop and two new edge states appear between the cells $i$ and $i + 1$. Subsequent hybridization between the now four edge states lifts the degeneracy. The "edge" states around $i$ are not robust against the small local perturbation of $w_i = 0$ in contrast to the real boundary edge states that require a non-local perturbation connecting the two ends. Hence, the system with one $w_i = 0$ still belongs to the regular topological phase of the SSH model. The value of $S^D$ is lower, however.

When two zero $v$'s or $w$'s are too close to each other, the approximation breaks down. More generally, when the disorder is too strong, it induces a phase transition beyond which no zero modes may exist anymore.

# 4   $S^D$ within a phase and scaling analysis at the phase transition

In this section and the next, we employ free fermion techniques to obtain $S^D$ for generic parameters of the system. This method is equivalent to exact diagonalization and relies on the fact that the SSH model describes non-interacting fermions. We briefly review this technique in Sec. 4.1. In Sec. 4.2, we obtain $S^D$ for a range of parameters around the phase transition where $S^D$ displays a system-size scaling behavior.

## 4.1   Computing the entanglement spectra

The combination of analytical and numerical techniques reviewed in Ref. [61,62] allows direct access to the spectrum and eigenvalues of any quadratic Hamiltonians. It computes efficiently

the reduced density matrix' entanglement spectra that are necessary to deduce $S^D$ in Eq. 5. The technique is faster than direct exact diagonalization as its complexity grows only algebraically in system size. The reader may also refer to Ref. [49] that also uses the same technique for the SSH model to study entanglement at the phase transition. The scaling analysis realized in Ref. [49] concerns the bipartite entanglement for increasing subset size for infinite or semi-infinite chains. In contrast, we use the technique in Sec. 4.2 to study $S^D$ for increasing system size.

The correlation matrix is related to the reduced density matrix. The many-body reduced density matrix $\rho_X$ of a subsystem $X$ can be written as

$$\rho_X = Z_X^{-1} e^{-\mathcal{H}_X}, \tag{16}$$

with $Z_X = \text{Tr}_X \left[ e^{-\mathcal{H}_X} \right]$ and where $\mathcal{H}_X$ is the *entanglement Hamiltonian* of $\rho_X$. $\mathcal{H}_X$ is a quadratic Hamiltonian as long as the system's Hamiltonian describes non-interacting fermions. The SSH Hamiltonian Eq. 1 preserves the number of particles, so the technique provides the eigenvalues of $\mathcal{H}_X$, i.e., the entanglement spectrum, using only the correlation matrix $(C_X)_{mn} = \langle c_m^\dagger c_n \rangle$ of the state of interest, where $m, n$ are site indices belonging to $X$. Indeed, the entanglement Hamiltonian and the correlation matrix are related [61]:

$$\mathcal{H}_X = \log \frac{1 - C_X}{C_X}. \tag{17}$$

The one-body eigenstates of the initial Hamiltonian $H$, the $\{\Phi_k\}_k$, are obtained with exact diagonalization (restricted to one-body states). The many-body ground state correlation matrix $C_X$ follows:

$$(C_X)_{mn} = \sum_{|k| < k_F} \Phi_k^*(m) \Phi_k(n), \tag{18}$$

where $k_F$ is Fermi's momentum. We then numerically diagonalize the matrix Eq. 18 (of the same length of $X$) using a computer. From the spectrum, we compute the entanglement entropies in $S^D$ using Eqs. 17, 16 and 4.

The procedure is also useful to track a time-dependent state and its entanglement: Starting from the ground state of $H_0 = \sum_{ij} h_{ij}^0 c_i^\dagger c_j$ for $t < 0$, the system evolves after a sudden quench at $t = 0$. The Hamiltonian becomes $H = \sum_{ij} h_{ij} c_i^\dagger c_j$ for $t > 0$. Along with the state $\rho(t)$, the correlation matrix acquires a time dependence:

$$\begin{aligned}
(C_X)_{mn}(t) &= \text{Tr}\left[ \rho(t) c_m^\dagger(0) c_n(0) \right] \\
&= \text{Tr}\left[ \rho(0) \, c_m^\dagger(t) c_n(t) \right] \\
&= \sum_{kk', m'n'} \Phi_k^*(m) \Phi_{k'}(n) e^{-iE_{k'} t} e^{iE_k t} (C_X)_{m'n'}(0) \Phi_{k'}^*(n') \Phi_k(m'),
\end{aligned} \tag{19}$$

where $E_k$ and $\Phi_k$ are the eigenvalues and the eigenvectors of $H$, $c_m^\dagger(t)$ (resp. $c_m(t)$) is the Heisenberg representation of $c_m^\dagger$ (resp. $c_m$), $(C_X)_{mn}(0)$ follows Eq. 18 for the ground state of $H_0$, and the sum over the $k$ and $k'$ in Eq. 19 includes all the momenta. Eq. 19 is valid for any reduced subsystem $X$. From the spectrum of $C_X(t)$ for any $X = A, B, A \cup B,$ and $A \cap B$, we obtain $S^D(t)$.

## 4.2 Phase diagram and scaling analysis

Using this technique, we compute $S^D$ and recover the expected phase diagram for the SSH model with open boundary condition in Fig. 2a). $S^D$ fulfils its role as a "topological detector" as it is non-zero in the topological phase ($v/w < 1$) and zero in the topological-trivial phase

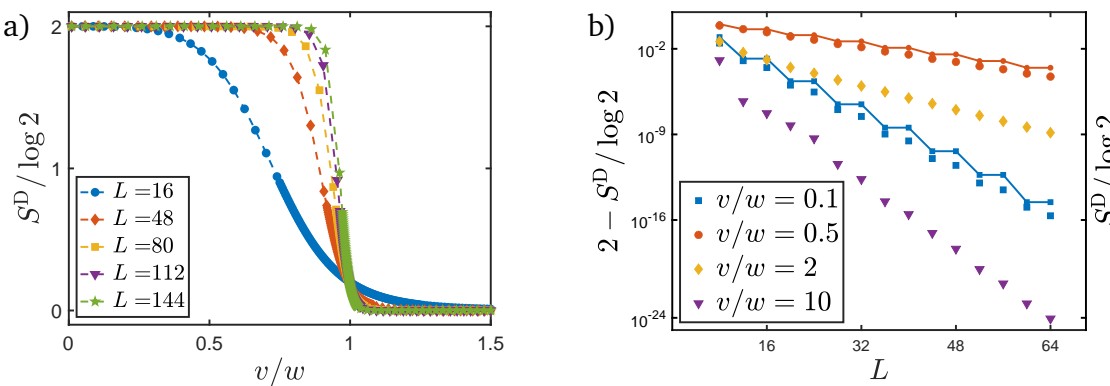

Figure 2: (Color online) a) $S^{\mathrm{D}}$ as a function of the ratio $v/w$ and the total length $L$ for $L = 2L_A = 2L_B = 4L_D$. The critical point is at $v/w = 1$. $S^{\mathrm{D}}$ is non-zero for the topological phase, and zero outside: $S^{\mathrm{D}}$ qualifies as a good topological detector. b) Scaling behavior of $S^{\mathrm{D}}$ towards its quantized convergence value. For the topological phase (left $y$-axis; $v/w = 0.1$ and 0.5 resp. squares and dots), $S^{\mathrm{D}}$ converges to $2\log 2$. For the non-topological phase (right $y$-axis; $v/w = 2$ and 10 resp. diamonds and triangles), $S^{\mathrm{D}}$ converges to zero. The increments on both the left and right $y$-axis are the same. The scaling behavior of $S^{\mathrm{D}}$ is exponential with the size of the chain $L = 2L_A = 2L_B = 4L_D$ for parameters in both phases. The results of the simulations (scattered points) agree with the analytic approximations of Sec. 3 (full lines). The saw-teeth variations of the latter are due to the alternating sign in Eqs. 14a and 14b.

($v/w > 1$). $S_2^{\mathrm{D}}$ (as in Eq. 11) is found identical, up to minor quantitative changes close to the phase transition.

Both the correlation length and the localization length increase close to the transition. The resulting spreading of the internal dimers and the edge states prevents a clean extraction of the edge entanglement, damping the value of $S^{\mathrm{D}}$. In the large size limit ($L_A, L_B, L_D \rightarrow \infty$), the well-quantized plateau of $S^{\mathrm{D}} = 2\log 2$ and $S^{\mathrm{D}} = 0$ extends over their whole respective phases according to the scaling of Fig. 2b).

We observe a system-size scaling behavior for $S^{\mathrm{D}}$ at the second-order phase transition as in Ref. [31]. We use the following Ansatz, typical of an order parameter:

$$S^{\mathrm{D}} L^{\frac{a}{b}} = \lambda \left( L^{\frac{1}{b}} (\alpha - \alpha_c) \right), \tag{20}$$

with fixed $L_A$ and $L_B$ so that $L(L_D)$ is the only scaling parameter left. $\alpha = v/w$ is the varying parameter and $\alpha_c$ is the critical value of this parameter at the phase transition. $\lambda(x)$ is the universal function at the phase transition, and $a$ and $b$ are entanglement critical exponents, similar to $\beta$ and $\nu$ for the 1D Ising chain at the paramagnetic/ferromagnetic phase transition. $\lambda(x)$ behaves asymptotically as:

$$\lambda(x) \rightarrow \infty \qquad\qquad \text{when } x \rightarrow -\infty, \tag{21a}$$
$$\lambda(x) \rightarrow 0 \qquad\qquad \text{when } x \rightarrow +\infty. \tag{21b}$$

The curve intersection and curve collapse of Figs. 3 give the value $\alpha_c = 0.958$, $a = 1.01$, and $b = 0.81$ for the best mean square fit. The exponents $a$ and $b$ are obtained as the optimal values from a discrete mesh of spacing 0.01. It is not straightforward to assign a rigorous interval of confidence to the values we have obtained. From the data shown in the inset of Fig. 3a, one can observe a drift of order 1% in the crossing position between the curves representing the two

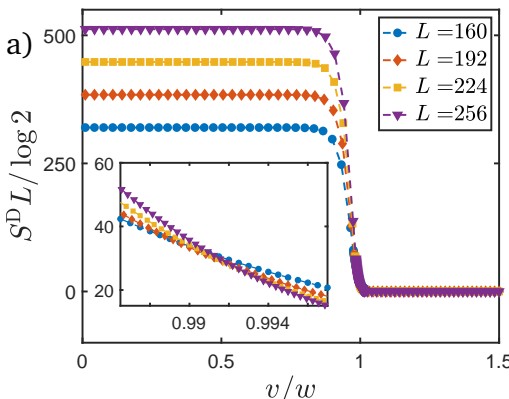
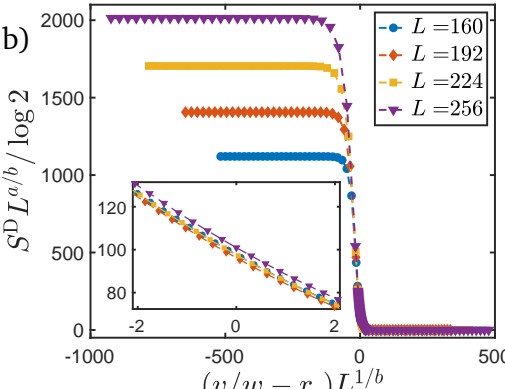

Figure 3: (Color online) a) Curve intersection of $S^D L$ as a function of the ratio $v/w$ for $L_A = L_B = 64$. It extracts the critical point at the crossing, $v/w = 0.958$ here (1 theoretically). b) Best curve collapse of the Ansatz Eq. 22 obtained for $a = 1.01$ and $b = 0.81$. The clear collapse in inset signals the universal behavior of $\lambda$ at the transitions.

smaller (blue and red line) and the two larger (yellow and violet) system sizes, respectively. It is thus reasonable to assume that the relative error on $\alpha_c$ is at the percent level.

The Ansatz Eq. 20 fails to describe the scaling behavior of $S^D$ close to quantized plateau at $2\log(2)$ at $\alpha = \alpha_t(L)$. $\alpha_t(L)$ gives an estimate of the transition region. With $L_A$ and $L_B$ still fixed, we have, in general:

$$S^D L^{\frac{a}{b}} = \Theta\left(L, L^{\frac{1}{b}}(\alpha - \alpha_c)\right), \tag{22}$$

such that, according to Fig. 3b:

$$\Theta\left(L, L^{\frac{1}{b}}(\alpha - \alpha_c)\right) = \begin{cases} L\log 2 & \text{when } L^{\frac{1}{b}}(\alpha - \alpha_c) < x^*(L), \\ \lambda\left(L^{\frac{1}{b}}(\alpha - \alpha_c)\right) & \text{when } L^{\frac{1}{b}}(\alpha - \alpha_c) > x^*(L), \\ \begin{array}{l} \text{a non universal} \\ \text{regularization} \end{array} & \text{when } L^{\frac{1}{b}}(\alpha - \alpha_c) \sim x^*(L), \end{cases} \tag{23}$$

where $x^*(L) = L^{\frac{1}{b}}(\alpha_t(L) - \alpha_c) < 0$ marks the end of the plateau and the start of the universal regime. We find neither $x^*(L)$ nor $\alpha_t(L)$ to be universal, a result that is not unexpected as the plateau is due to a UV property of the phase.

# 5 Invariance of $S^D$ after global quenches

In this section, we show that disconnected entanglement carries robust signatures of quantization after global quantum quenches, as expected for topological invariants associated with particle-hole symmetry (PHS) [40].

In Sec. 5.1, we elaborate on two arguments discussed in Ref. [40]. These arguments predict the conservation of topological edge states over time, and thus, according to Eq. 8, the conservation of $S^D$. This prediction is consistent with our simulations in Sec. 5.2. Quantum quenches are thus ideal test-bed to determine whether a given quantity is indeed a topological invariant associated with particle-hole symmetry.

## 5.1 Invariance of $S^{\mathrm{D}}$ during unitary evolution: the role of particle-hole symmetry

We provide here the two arguments explaining why topological invariants associated with PHS are invariant to symmetry-preserving quenches that are presented in Ref. [40]. In particular, we explicitly prove why the D classification applies to the Hamiltonian in the interaction picture, which is an important step in one of the two arguments. We then use one of the conclusion of Ref. [40]: the stability of the edge states after a quench, to prove the invariance of $S^{\mathrm{D}}$. Thus, we argue that $S^{\mathrm{D}}$ behaves like a topological invariant that keeps track of the initial maximally entangled edge states.

Given a system, a quantized topological invariant associated with a symmetry is computed from the ground state. If this state locally evolves in time without breaking the symmetry, the topological invariant remains constant. To prove the statement for PHS, we denote as $\mathcal{H}^i$ and $\mathcal{H}^f$ the initial Hamiltonian (before quench) and the final Hamiltonian (after quench), respectively. If $|\psi(0)\rangle$ is a ground state of $\mathcal{H}^i$, then the state evolves after the quench as $|\psi(t)\rangle = U(t)|\psi(0)\rangle$, where the unitary evolution operator depends only on $\mathcal{H}^f$ and $t$. $\mathcal{H}^i$ is PHS if $CH^*C^\dagger = -H$, where $C$ is the PHS operator and $\mathcal{H}^i = \sum_{ml} \psi_m^\dagger H_{ml} \psi_l$. In that case, $\rho(0) = |\psi(0)\rangle\langle\psi(0)|$ is also PHS, i.e., $C\rho^*C^\dagger = 1 - \rho$. If $\mathcal{H}^f$ is also PHS, so will $\rho(t)$. So if a topological invariant associated with PHS is initially fixed to a quantized value by $\rho(0)$, this value remains quantized along the PHS-preserving dynamics.

The second alternative argument consists in viewing the evolving state as the ground state of the quenched Hamiltonian in the interaction picture. Time becomes a parameter of this fictitious Hamiltonian on which we apply the topological insulator classification associated with the PHS symmetry. Specifically, $|\psi(t)\rangle$ is a ground state of the fictitious Hamiltonian $\mathcal{H}^{\mathrm{fic}}(t)$:

$$\mathcal{H}^{\mathrm{fic}}(t) = U(t)\mathcal{H}^i U(t)^\dagger.$$

$\mathcal{H}^{\mathrm{fic}}(t)$ is PHS if $\mathcal{H}^i$ and $\mathcal{H}^f$ are PHS. The spectrum is invariant in time, so there is no gap closing along the dynamics. Since $\mathcal{H}^i$ and $\mathcal{H}^f$ are finite-ranged, we show explicitly that $\mathcal{H}^{\mathrm{fic}}(t)$ is short-ranged. Indeed, using the Baker-Campbell-Hausdorff formula,

$$\mathcal{H}^{\mathrm{fic}}(t) = H^i + \sum_{n=1}^{\infty} \frac{(it)^n}{n!} C_n(H^f, H^i), \tag{24}$$

where $C_n(H^f, H^i) = [H^f, [H^f, ...[H^f, H^i]...]]$ where $H^f$ appears $n$ times and $[A, B]$ is the commutator between $A$ and $B$. Assuming that both $H^i$ and $H^f$ only involve nearest neighbors hoping, we write $n = 2l$ when $n$ is even, $n = 2l - 1$ when n is odd. Thus:

$$C_{2l}(H^f, H^i) = \sum_{k=0,l} \alpha_{i,k}(l) c_i^\dagger c_{i+2k+1} + H.c. \tag{25a}$$

$$C_{2l-1}(H^f, H^i) = \sum_{k=1,l} \tilde{\alpha}_{i,k}(l) c_i^\dagger c_{i+2k} + H.c., \tag{25b}$$

where $|\alpha_{i,k}| \leq \Lambda^{2l+1} 2^{2l} S_k^l$ and $|\tilde{\alpha}_{i,k}| \leq \Lambda^{2l} 2^{2l-1} S_k^l$ (for all $i$). $\Lambda$ is the largest absolute value of all the hopping amplitude in $H^i$ and $H^f$ and $S_k^l$ are obtained from the Catalan triangle [63] such that ($l \in \mathbb{N}^*$, $0 \leq k \leq l$):

$$S_k^l = \mathrm{Binomial}(2l, l-k) - \mathrm{Binomial}(2l, l-k-1) \quad \text{if } n \text{ is even}, \tag{26a}$$

$$S_k^l = \mathrm{Binomial}(2l-1, l-k) - \mathrm{Binomial}(2l-1, l-k-1) \quad \text{if } n \text{ is odd}, \tag{26b}$$

where by convention Binomial$(n, -1)=0$. Rewriting $\mathcal{H}^{\text{fic}}(t)$ as:

$$\mathcal{H}^{\text{fic}}(t) = \sum_{i,r} \beta_{i,r}(t) \left( c_i^\dagger c_{i+r} + H.c \right).$$

We thus have:

$$|\beta_{i,r}(t)| \leq \sum_{l=k}^{\infty} \Lambda^{2l+1} \frac{(2t)^{2l}}{(2l)!} S_k^l \quad \text{if } r = 2k-1 \text{ using Eq. 26a,} \tag{27a}$$

$$|\beta_{i,r}(t)| \leq \sum_{l=k}^{\infty} \Lambda^{2l} \frac{(2t)^{2l-1}}{(2l-1)!} S_k^l \quad \text{if } r = 2k \text{ using Eq. 26b.} \tag{27b}$$

Using *Mathematica*:

$$\sum_{l=k}^{\infty} \Lambda^{2l+1} \frac{(2t)^{2l}}{(2l)!} S_k^l = \frac{(2k+1) I_{2k+1}(4\Lambda t)}{2t} = o(1/k), \tag{28a}$$

$$\sum_{l=k}^{\infty} \Lambda^{2l} \frac{(2t)^{2l-1}}{(2l-1)!} S_k^l = \frac{2k I_{2k}(4\Lambda t)}{2t} = o(1/k), \tag{28b}$$

where $I_n(x)$ is the modified Bessel function of the first kind such that $I_0(0) = 1$. Thus, $\beta_{i,r}(t) = o(1/r)$ on all sites and for all times: although the range of $\mathcal{H}^{\text{fic}}(t)$ increases with time, the Hamiltonian is short range.

$\mathcal{H}^i$ and $\mathcal{H}^{\text{fic}}(t)$ are thus connected unitarily, continuously, locally and without closing the gap, as if by an adiabatic connection [64–66] (no extra hypothesis of adiabaticity was imposed in the reasoning). Therefore, unless the system experiences a dynamical phase transition, $\mathcal{H}^i$ and $\mathcal{H}^{\text{fic}}(t)$ are in the same topological phase relative to the PHS, and if one has robust edge states, so does the other. Consequently, the topological invariant associated with the PHS (like the Zak phase modulo $2\pi$) is also invariant. The reasoning extends to interacting systems according to Ref. [40]. Note that, when the system is finite, and after a certain time $t \sim t_c$, $U(t)$ effectively becomes an infinite-ranged (non-local) transformation such that the topology associated with the state $|\psi(t)\rangle$ is not well-defined anymore. The topological invariant starts then to vary.

We stress that these considerations concern PHS (a unitary symmetry leaving time invariant) only, and not the time-reversal (TRS) or chirality symmetry (CS), which are broken by time evolution in this reasoning. Hence, a topological invariant fixed by, e.g., TR, may vary in time even when both $\mathcal{H}^i$ and $\mathcal{H}^f$ are TRS. This means that, while there is an infinite amount of topological phases in the BDI class that can be distinguished using the Zak phase (in $\pi\mathbb{Z}$, fixed by TRS, PHS, and CS), only two are distinguished by the Zak phase modulo $2\pi$ (fixed by THS) as time evolves.

Thus, if $S^D$ is a topological invariant, then $S^D$ of a topological ground-state (at equilibrium) is conserved when the state is time-evolved by any local, unitary, and symmetry-preserving operator until a time $t_c$ fixed by the finite size of the chain. We argue that the quenches we consider only induce such time-evolution. The conservation of the topological invariants of the system implies the conservation of the associated bulk-boundary correspondence. The conserved correspondence implies conserved edge states, and, thus, conserved $S^D$. Indeed, we observe this conservation of $S^D$ in Sec. 5.2. We conclude that $S^D$ is likely a topological invariant.

The reasoning also shows that $S^D$ keeps track of the topology of the initial state while the topological invariant associated with PHS, the Zak phase (modulo $2\pi$), does not. Indeed, the Zak phase in the topological phase of the SSH model is $2\pi = 0$ modulo $2\pi$ (while the Zak

phase of, e.g., the Kitaev model is $\pi$). Hence, the topological SSH phase and the trivial phase are not expected to be topologically different in the dynamical context. Yet, $S^{\mathrm{D}}$ distinguishes between states with and without long-range edge entanglement, an observable feature that we traced back to topology.

We conclude that $S^{\mathrm{D}}$ is likely a topological invariant. Furthermore, $S^{\mathrm{D}}$ keeps track of the topology of the initial state, whereas another topological invariant like the Zak phase modulo $2\pi$ does not.

## 5.2   Invariance of $S^{\mathrm{D}}$ after quenches: finite-size scaling analysis

We performed an extensive investigation of the evolution of $S^{\mathrm{D}}$ after a quantum quench within and across the topological phase. We used the procedure detailed in Sec. 4.1. Specifically, we first derive the one-body eigenvalues of the desired initial Hamiltonian $H_0$. Using these eigenvalues and Eq. 18, we obtain the full correlation matrix $C_X(0)$ of the initial ground state. Similarly, we derive both the one-body spectrum and the eigenvalues of the Hamiltonian post-quench $H$. Using $(C_X)_{mn}(0)$, the spectrum and eigenvalues of $H$, and Eq. 19, we obtain the time-dependent full correlation matrix $C_X(t)$ of the quenched state. From $C_X(t)$, we finally compute $S^{\mathrm{D}}(t)$ like in the static case. Fig. 4a) gives the representative example of the time evolution of $S^{\mathrm{D}}(t)$ after a quench from the topological phase to the trivial phase. Instead, Fig. 4c) corresponds to a quench from to trivial phase to the topological phase.

For both quenches and for quenches within the topological or within the trivial phase, we observe the same phenomenology: $S^{\mathrm{D}}$ sticks to its initial value until a certain timescale $t_c$ that depends on the quench and the size of the system as in Ref. [31]. The corresponding phenomenon in the bipartite entanglement entropy is a constant offset during the time evolution [67]. We define the timescale $t_c$ as the time when $S^{\mathrm{D}}$ varies of $2\log 2/100$ from its initial value (dotted line in inset of Fig. 4a) and c)). We observe that $t_c \sim L/\eta$ when $L > 100$ in Fig. 4b) and d). $\eta$ increases when the amplitude of the quench increases. When $L \to \infty$, $t_c \to \infty$ showing that $S^{\mathrm{D}}$ behaves like a topological invariant.

# 6   Robustness of $S^{\mathrm{D}}$ to disorder

For 1D non-interacting systems, Anderson localization kicks in as soon as disorder is introduced [68] (or reviewed in Ref. [69]). This localization is not antagonistic to topological phases. Both can coexist. The disorder can even favor the topological phase, as known for the case of quantum Hall effects in $D > 1$. In the SSH model, disorder can extend the topological phase past $v/w > 1$. This extended regime is called a topological Anderson insulator [70–72].

Using $S^{\mathrm{D}}$ we successfully reproduce the disorder-induced phase diagram of the SSH model, see Fig. 5. This phase diagram is known and was partially measured for uniform disorder [46] and is known for quasiperiodic potential [73]. The former work extrapolated the winding number from measurements. This topological invariant stays well quantized to 1 or 0 (mod 2) despite the disorder for both the topological and the trivial phase. We observe a similar behavior for $S^{\mathrm{D}}$. The robustness to disorder of $S^{\mathrm{D}}$ follows the robustness of the edge states of SPTP [74].

Specifically, we consider uniform, chirality-preserving disorder on the hoppings of Eq. 12:

$$w_i = w + W_1 \delta_i, \tag{29a}$$

$$v_i = v + W_2 \Delta_i, \tag{29b}$$

with $w = 1$ fixed, $\delta_i$ and $\Delta_i$ being random variables of uniform distribution in $[-0.5, 0.5]$. We then average $S^{\mathrm{D}}$ over the realizations. For weak disorder, Fig. 5 ($2W_1 = W_2 = W$) shows that

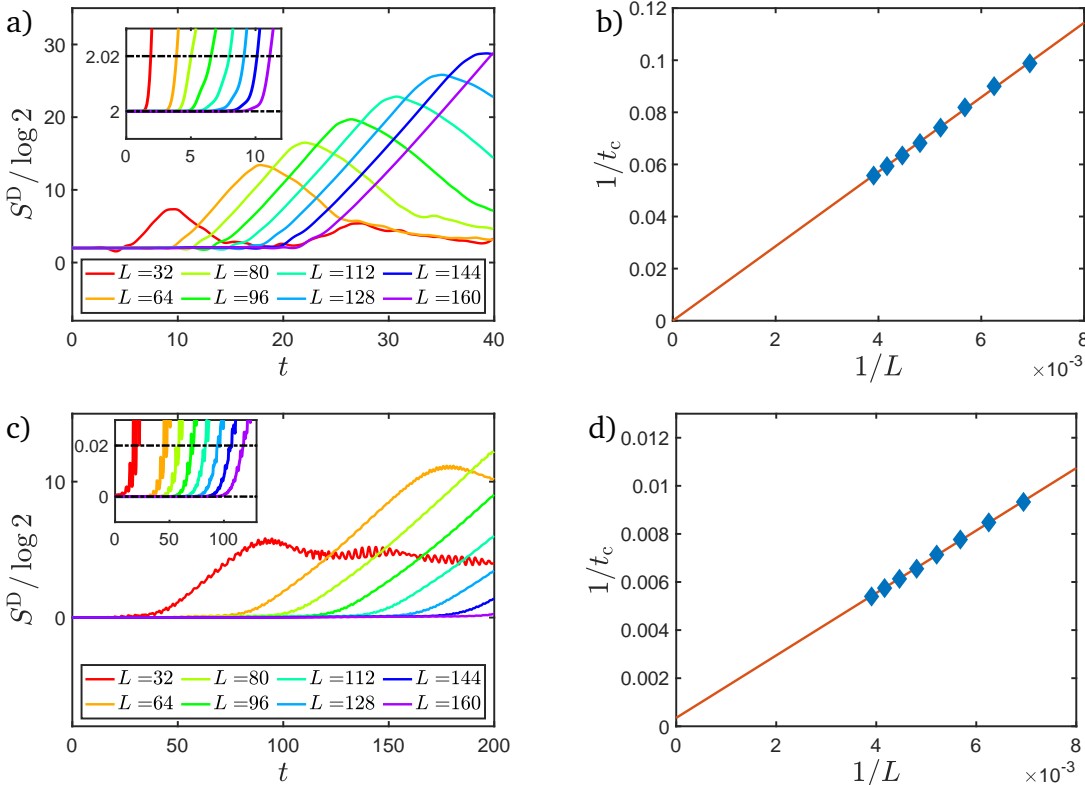

Figure 4: (Color online) a) and c) Time evolution of $S^D$ after quenching the Hamiltonian from a) $v/w = 0.1$ (topological phase) to $v/w = 1.5$ (trivial phase) and c) $v/w = 1.5$ to $v/w = 0.1$ at $t = 0$ and for different total length $L$ with $L_A = L_B = 2L_D = L/2$. $S^D$ remains at its initial value until finite-size effects change it at $t \sim t_c$. Insets: zoom on the graph around $t \sim t_0$ and for $|S^D(t) - S^D(0)| \lesssim 2\log 2/100$. b) and d) Scaling behaviour of $t_c$ after the sudden quench a) and c) respectively. The point corresponding to smallest $L$ in both b) and d) is not included in the linear fit (orange line). In d), the fit includes the origin within two standard deviation. Thus, when $L \to \infty$, $t_c$ diverges, demonstrating that $S^D$ does not evolve after unitary evolution for large systems.

the topological phase is stable when $S^D = 2\log 2$. The phase is trivial at strong disorder, and $S^D = 0$.

We also observe the topological Anderson insulator regime like in Ref. [46]. For $W_1 = W_2 = W$ (not shown), the phase transition line is monotonous with $W$. The locations of both transition lines we observe are compatible with the literature [46, 73]. Unlike the critical point $W_1 = W_2 = 0$ of Sec. 4.2, $S^D$ is well-quantized at either 0 or $2\log 2$ around the phase transition line. Its distribution is however a bimodal hence the damped value of the average at the transition. It is unclear to us if such a distribution is a marker of first order phase transition.

# 7 Disconnected entropies in the BDI class

We now discuss the generality of our results in the context of the BDI class of the tenfold-way [44, 75, 76].

The SSH can be mapped locally to a stack of two coupled Kitaev wires [77]. In the formalism of Ref. [78], this stack is called a 2-chain whereas the non-interacting Kitaev chain is

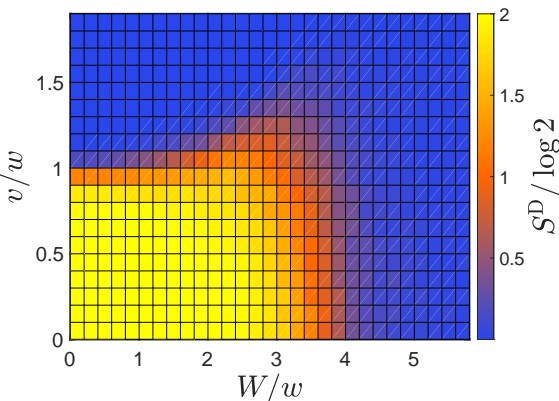

Figure 5: (Color online) Phase diagram of the SSH model obtained with $S^{\mathrm{D}}$ for the uniform disorder of Eq. 29 with $2W_1 = W_2 = W$. We set $L_A = L_B = 2L_D = 32$. $S^{\mathrm{D}}$ is averaged over 400 realizations for each point. The "bump" of the topological phase (in yellow) constitutes the topological Anderson insulator regime.

the 1-chain. Both the 1- and 2-chain can be understood with the same $\mathbb{Z}$ classification. The topological phases they display belong to different classes within the BDI classification as they have a different value for their topological invariant: the Zak phase. Specifically, the Zak phase is $\pi$ in the 1-chain and $2\pi$ in the 2-chain.

The edge states of the two chains are also different. Both chains have one (e.g., left) edge mode protected by the time-reversal symmetry, while the other (right) mode is protected by both the parity and the time-reversal symmetry [77]. In the 1-chain, the left and right parity operators do not commute. It is impossible to write the left and right edge modes as linearly independent in the same local basis. Thus, the ground state naturally requires a non-local description due to these edge modes. This specific form of non-locality implies a non-zero value for $S^{\mathrm{D}}$.

In the 2-chain, the left and right parity operators commute. The edge modes can be written independently from each other in a local basis (cf Sec. 3), and are local. For an infinite chain, the edge entanglement can thus be zero. As we showed in this paper, that is not the case, and the edge entanglement remains quantized and is maximal (given the Hilbert space dimension of the edge modes) for any finite chain.

We extend the conclusions shared between the 1-chain and the 2-chain to the whole BDI classification. Indeed, all topological phases of BDI either have non-local (fractional) edge states or local edge states, like in the 1- and the 2-chains respectively [77]. Only the association between the protecting symmetry (time-reversal or the composition parity and time-reversal) and the protected edge (left or right) varies. These variations should have no consequences to $S^{\mathrm{D}}$ that does not distinguish between left and right. We conclude that the validity of $S^{\mathrm{D}}$ extends to all BDI, as the phenomenology of edge modes is the same as the two models considered so far.

While this is not directly relevant for the model discussed here, the generality of our conclusion likely extends to the D classification. Even without time-reversal symmetry, the Kitaev wire displays a topological phase and a trivial phase. Both phases belong to the D classification which is a $\mathbb{Z}_2$ classification associated with the only particle-hole symmetry. The edge states of the topological Kitaev wire in the D class are the same as in the BDI class [8].

---

[8] although they are not protected by the same set of symmetries.

# 8 Conclusions

We have shown how entanglement entropies distinguish topological and non-topological insulating phases in the Su-Schrieffer-Heeger one-dimensional model with open boundary conditions. This entanglement is quantified by the disconnected entanglement entropy $S^D$ computed for the ground state of the system. It is 0 in the trivial phase and $2\log 2$ for the topological phase in the large system limit. We related $S^D$ to the number of zero-mode edge states, a topological invariant. Thus, $S^D$ is quantized and enjoys robustness disorder. As the model is particle-hole symmetric, $S^D$ is also invariant during local, unitary, and symmetry-preserving time evolution for large system size. $S^D$ also displays a universal scaling behavior when crossing the phase transition, akin to an order parameter. Numerical simulations show that modest and experimentally accessible partition sizes are sufficient for $S^D$ to reach its quantized regime. Finally, combining the present findings with older results on fermionic $S^D$ in the Kitaev chain, we argued that our conclusions extend to the full BDI class of the topological insulators and superconductors classification.

To complete the comparison of $S^D$ to a topological invariant, it would be interesting to investigate the evolution of $S^D$ when the protecting symmetry is explicitly broken and the maximal entanglement of the edge state is no more set topologically, such as in the Rice-Mele model [79]. It would also be interesting to use entanglement topological invariants to characterize the real-time dynamics of other instances of topological insulators in the presence of a bath [66].

# Acknowledgements

We are grateful to R. Fazio, G. Magnifico, T. Mendes-Santos, A. Scardicchio, and S. Taylor for useful discussions. The code used to obtain the data presented in the paper can be found in [80].

**Funding information** This work is partly supported by the ERC under grant number 758329 (AGEnTh), has received funding from the European Union's Horizon 2020 research and innovation programme under grant agreement No 817482, and from the Italian Ministry of Education under the FARE programme (R18HET5M5Y). This work has been carried out within the activities of TQT.

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
