# Peer review of "Topological entanglement properties of disconnected partitions in the Su-Schrieffer-Heeger model"

_SciPost Physics Core, doi:SciPost Phys. Core 3, 012 (2020)_

## Round 1 · Referee Report · Anonymous (Referee 1) · 2020-8-10

Report

The authors study the disconnected entanglement entropy in the SSH chain. Their results are in agreement with the well-known phase digram consisting of a phase with edge modes and one without (for the open chain). Thus I do not see anything that is not expected right from the outset. I rather view the manuscript as a case study of the disconnected entanglement entropy in another model. Thus I conclude that the SciPost Physics Core expectations are not met.

More generally I am also unsure whether the results really show that the studied disconnected entanglement entropy picks up information about the topological properties of the system. My problem with this is that the SSH chain doesn't have the same topological properties as, say, the Kitaev chain (which was already studied in Ref. 28). True, one of the phases of the SSH chain possesses edge modes, but these do not show the same degree of protection against local perturbations as in the Kitaev chain. Thus I do not think that from the results one can draw conclusions on topological properties in generality.

  • validity: -
  • significance: -
  • originality: -
  • clarity: -
  • formatting: -
  • grammar: -

Author:  Pierre Fromholz  on 2020-11-02  [id 1024]

(in reply to Report 1 on 2020-08-10)
Category:
answer to question
reply to objection

We thank the referee for their consideration of our work and their useful remarks. We identified three points of concern that we now reply to:

The referee writes: * I do not see anything that is not expected right from the outset. [...] the manuscript [is] a case study of the disconnected entanglement entropy in another model.

Our response: We argue that the systematic entanglement between topological edge states, while expected from a few examples, has never been rigorously proven. We give both theoretical and numerical proof of it for the SSH chain in the text. This conclusion could not be deduced from Ref.[28] (cf the second point below).

Furthermore, one of the main results of the article is the scaling analysis of $S^D$ at the topological phase transition (with and without disorder), and the determination of corresponding entanglement critical exponents. To the best of our knowledge, such a critical behavior for entanglement has never been predicted nor observed in models within the SSH universality class. Thus we politely, but strongly, disagree with the Referee, that such behavior could have been predicted from the outset. In support of the fact that such scaling behavior of topological contributions to entropies is vastly unexplored, we note that a recent work by Lavasani et al. (arXiv:2004.07243) has discussed similar scaling in the context of random unitary circuits, albeit in a situation (transition between area and volume law phases) that is clearly orthogonal to our case (which deals with ground states of local Hamiltonians).

We also show that unlike other topological invariants, e.g. the Zak phase modulo $2\pi$ relevant for the model, $S^D$ keeps track of the existence (or not) of edge states in the initial states, differentiating the topological and the trivial phase despite local and unitary dynamics.

Finally, while the manuscript does focus on the case of the SSH model, we argue that, together with Ref.[28] (now Ref. [31]), we can conclude for all topological insulators/superconductors in the BDI and D class (cf the third point). We now highlight more these main results in the introduction, the conclusion, and the titles of the sections.

The referee writes: * [...] the phases of the SSH chain possesses edge modes, but these do not show the same degree of protection against local perturbations as in the Kitaev chain [...].

Our response: We are not certain what the Referee meant here. Following, e.g., Verresen R. et al. Phys. Rev. B 96, 165124 (2017), the SSH chain can be mapped (locally) to the 2-Kitaev chain which is indeed different from the 1-chain, i.e., the non-interacting Kitaev chain, studied in Ref.[28]. Both the 1- and 2-chain can be understood with the same $\mathbb{Z}$ classification (the BDI class), although they belong to different classes.

Like the 1-chain, the 2-chain has its left mode protected by the time-reversal symmetry, while the right mode is protected by the parity and time-reversal together. Unlike the 1-chain, in the 2-chain, the left and right parity operators commute, meaning that the edge modes can be local. For an infinite chain, the edge entanglement can thus be zero. Yet, we observe and interpret that this entanglement remains and is maximal for a finite chain. This result could not be predicted by the study of the 1-chain of Ref.[28]. We reproduced this discussion in the text (Sec.7.1).

Beyond these considerations, it is worth emphasizing a much stronger element of difference with Ref.[28]. In the latter, the topological invariance of the disconnected entropy was understood utilizing a lattice gauge theory analogy. That analogy was enabled by the well-known direct correspondence between the Kitaev chain, and Wegner gauge theory. This analogy is not applicable to the case presented in our work.

The referee writes: * [...] from the results one can [not] draw conclusions on topological properties in generality.

Our response: We added Sec.7.1 to specifically discuss this point and justify the extension of our claims to the full BDI and D classes only. Indeed, the association of the results of both Ref.[28] and the present manuscript allows us first to extend the conclusion to the whole BDI classification. Following the previous discussion on the commutation of the left and right parity operators, we distinguish two types of symmetry-protected topological phases. First, phases with non-local edge modes, i.e. phases for which it is impossible to write in the same local basis independent left and right edge modes. This non-locality due to the edges present in the ground state is precisely what explains the quantized value of $S^D$ in the 1-chain of Ref.[28]. The second type of phase have local edge modes, like in the SSH model. Yet, we argue in the present manuscript that the ground state still involves long-range entanglement due to the edges as long as the chain is finite. Despite involving a different mechanism, all the conclusions on $S^D$ in Ref.[28] extend to the SSH chain. Following, e.g., Table 1 of [Verresen R. et al. Phys. Rev. B 96, 165124 (2017)], all of the topological phases in the BDI class fall into one of these two categories. We thus reasonably think that the validity of $S^D$ extends to all the topological phases of BDI, as only their edge properties matter for $S^D$. It could also extend beyond the classification of ground states onto steady states, as the aforementioned results in the context of random unitary circuits seems to suggest.

The conclusion also extends to the D classes: even without the time-reversal symmetry, the Kitaev wire still displays a topological phase. The phase then belongs to the D class with a $\mathbb{Z}_2$ classification (protected by parity or particle-hole symmetry). In this class, the edge states of the model are the same as the BDI class (although they are not protected by the same set of symmetries.), thus the conclusions of Ref.[28] on $S^D$ remain valid.

We hope that, with these comments and the corresponding changes applied to the text in responding to the Referee's critiques, we have clarified the novel aspect of our findings.

---

## Round 1 · Referee Report · Anonymous (Referee 2) · 2020-8-12

Strengths

The paper is clearly written and presented, with all necessary definitions included and calculations explained.

Weaknesses

The paper presents only an incremental advance or variation on what the authors have studied and published elsewhere.

Report

In the paper "Topological entanglement properties of disconnected partitions in the Su-Schrieffer-Heeger model" by Micallo et al they study the disconnected entanglement entropy for the SSH model. The disconnected entanglement entropy is proposed as a necessary and sufficient condition for diagnosing the topological invariant. In this work the authors SSH model. Both disorder and dynamics following a quench are also studied. In general the paper appears to be technically sound and clearly written, although there are several important references missing, and there are several statements which require clarification. I will detail these below. The results are all clearly argued and explained with all relevant information given.

As this paper applies an established technique to an already very well studied toy model, it does not appear to satisfy the publication requirements of SciPost Physics Core, and would be more suitable for a more specialized journal. Indeed some of the authors, in Ref. 28, already applied these techniques to the very closely related 1D Kitaev model, including both disorder and interaction effects. As the Kitaev model can be mapped to the SSH model in part of its parameter space it is hard to see what new information is gained here at all. I understand that using the disconnected entropy as a sign of topology is a conjecture, and as such more numerical evidence is welcome for this problem, but considering the work already in Ref. [28] I do not think this is sufficient for the publication requirements of SciPost Physics Core. The authors would need to argue more clearly how this work satisfies these requirements.

Requested changes

1. The following references which deal with entanglement entropy in the SSH model are missing and seem directly relevant to me, as all of these are discussed in this paper at various points.
S. Ryu and Y. Hatsugai, ‘Entanglement Entropy and the Berry Phase in the Solid State’, Physical Review B 73, (2006) 245115, https://doi.org/10.1103/PhysRevB.73.245115. - To the best of my knowledge this is the first calculation of the entanglement entropy for the SSH model.
L. Campos Venuti, C. Degli Esposti Boschi, and M. Roncaglia, ‘Long-Distance Entanglement in Spin Systems’, Physical Review Letters 96, (2006): 247206, https://doi.org/10.1103/PhysRevLett.96.247206.
L. Campos Venuti et al., ‘Long-Distance Entanglement and Quantum Teleportation in XX Spin Chains’, Physical Review A 76, (2007) 052328, https://doi.org/10.1103/PhysRevA.76.052328. - I believe these are the first papers which discusses the long range entanglement which originates from edge states.
J. Sirker et al., ‘Boundary Fidelity and Entanglement in the Symmetry Protected Topological Phase of the SSH Model’, Journal of Statistical Mechanics: Theory and Experiment 2014, (2014) P10032, https://doi.org/10.1088/1742-5468/2014/10/P10032. - This paper contains a detailed analysis of the scaling of the entanglement entropy for the SSH chain.
N. Sedlmayr et al., ‘Bulk-Boundary Correspondence for Dynamical Phase Transitions in One-Dimensional Topological Insulators and Superconductors’, Physical Review B 97, (2018) 064304, https://doi.org/10.1103/PhysRevB.97.064304. - This paper considers the dynamics of the entanglement entropy for the SSH chain following a quench.

2. In the abstract it is written that "To corroborate the topological origin of the quantized values of SD, we show how the latter remain quantized after applying unitary time evolution in the form of a quantum quench, a characteristic feature of topological invariants."

I would question that this is a characteristic feature of a topological invariant. For example, if I take the usual invariant for a 1D BDI model, the Zak-Berry phase, it is a property of the ground state. Following a quench the system is very far from its ground state and there is no meaning to use this invariant. It seems to me that this has been mixed up with the definition fo a topological phase which states that all ground states joined by (symmetry preserving) adiabatic unitary evolution (without closing the gap), are topologically equivalent. However it is not time evolution that is referred to here. (See for example Ref [9].)

It is also not clear to me that this is a desirable property of a topological invariant. Following a quench the system is far from equilibrium, why should one believe that the topological properties of its ground state are preserved? Nonetheless I agree that this is an interesting observation for $S^D$, I would appreciate more discussion of this point. (See also further related points below.)

3. In the first paragraph on p2 the authors write that the TEE "works both in and out of equilibrium". However as afar as I can see tracing the references back, in Hamma et al. (‘Entanglement, Fidelity, and Topological Entropy in a Quantum Phase Transition to Topological Order’, Physical Review B 77, (2008) 155111, https://doi.org/10.1103/PhysRevB.77.155111.) which is not referenced directly, they only deal with adiabatic evolution, not quenches or generic non-equilibrium scenarios. In Ref. [6] they are not dealing with the TEE but the usual EE as far as I can see, they they do consider quenches. Could the authors provide a clear reference for this statement?

4. In the footnote on p2 the authors write that the entanglement spectrum for the Ising and Kitaev model are the same, although only one of them is topological. Do the authors know a counterexample which is not mappable to a topological model?

5. At the end of p3 it should say $v/w/gg 1$ for the condition when sites within a unit cell dimerized. For $v/w$ close to 1 coupling both within and between unit cells is similar.

6. On p4 the authors write that the chiral $S$ operator is anti-unitary. In fact it is $T$ and $C$ which are anti-unitary, not $S$.

7. Note that relevant to section 3, an semi analytical solution exists for the SSH model, which allows some analytical calculations to be performed. See Byeong Chun Shin, ‘A Formula for Eigenpairs of Certain Symmetric Tridiagonal Matrices’, Bulletin of the Australian Mathematical Society 55 (1997) 249, https://doi.org/10.1017/S0004972700033918.

8. I would appreciate an equivalent of figure 4 for the opposite quench case, to demonstrate the difference. Also, the inset of figure 4a is not explained in the caption.

  • validity: high
  • significance: ok
  • originality: low
  • clarity: high
  • formatting: excellent
  • grammar: good

Author:  Pierre Fromholz  on 2020-11-02  [id 1025]

(in reply to Report 2 on 2020-08-12)

We thank the referee for their interest in our work, their reading, and their useful comments. We reply to the two main points of concern we identified, followed by their numbered suggestions.

The referee writes: * [...] this paper applies an established technique to an already very well studied toy model, [so] it does not appear to satisfy the publication requirements of SciPost Physics Core

Our response: The paper's originality lies in its main results: the topological protection of $S^D$ for a symmetry-protected topological phase with fermionic edge states (not Majorana) and the scaling analysis of $S^D$ at the topological phase transition with the determination of its corresponding critical entanglement exponents. We also show that $S^D$ behaves dynamically like a topological invariant and that unlike other topological invariant (such as the Zak phase modulo $2\pi$ relevant for the model), $S^D$ keeps track of the existence (or not) of edge states in the initial states. We now stress more these results in the introduction, the conclusion, and the titles of the sections.

The referee writes: * As the Kitaev model can be mapped to the SSH model in part of its parameter space it is hard to see what new information is gained here at all.

Our response: We agree with the referee that the SSH chain can be mapped to a Kitaev model which does provide an educated guess for the quantized value of $S^D$. This Kitaev model, however, is not the chain studied in Ref.[28]. Taking, e.g. the definition of [Verresen, Moessner, and Pollmann, One-dimensional symmetry protected topological phases and their transitions, Phys. Rev. B 96, 165124 (2017)], the SSH chain can be mapped (locally) to a 2-Kitaev chain, while the chain in Ref.[28] is a 1-chain only. This means that deep in the topological phase $v \ll w$, the SSH model is equivalent to two decoupled Kitaev chain. In that case, indeed, we expect that the entanglement probed at each cut is double compared to a single Kitaev chain. In particular, we expect that $S^D$ of SSH is double the $S^D$ of the Kitaev wire, which we observe exactly. This value remains constant in the whole phase as we argue that $S^D$ is linked to a topological invariant.

The SSH chain, however, does not display the same topological phase as the Kitaev wire. It also has the continuous U(1) symmetry, absent for the 1-chain. Thus, the topological phase transition is very different. To compare with [28], one of the main questions this article answers is: is the universality (scaling analysis and topological critical exponents) of $S^D$ observed in [28] at the one topological phase transition of the 1-chain a generic feature? While suggested in [28], it was impossible to answer then. The present article conclusively answer that universality at the transition is a generic feature, and that the observed critical exponents depend on the phase transition and are not straightforwardly deducible from the 1-chain case.

Finally, we argue that the time invariance of a non-zero $S^D$ to quenches does not follow Ref.[28], cf our reply to point 2 below. We modified the introduction and the discussion in conclusion to make this point more explicit. In particular, the new Sec.7.1 details how different the two chains are.

The referee writes: 1. The following references which deal with entanglement entropy in the SSH model are missing and seem directly relevant to me, as all of these are discussed in this paper at various points. S. Ryu [et al.] [...]. - To the best of my knowledge this is the first calculation of the entanglement entropy for the SSH model. L. Campos Venuti [et al] [...] [and] L. Campos Venuti et al. [...] - I believe these are the first papers which discusses the long range entanglement which originates from edge states. J. Sirker et al.[...] - This paper contains a detailed analysis of the scaling of the entanglement entropy for the SSH chain. N. Sedlmayr et al.[...] - This paper considers the dynamics of the entanglement entropy for the SSH chain following a quench.

Our response: 1. We thank the Referee for their rich and on-point suggestions of references that we were not aware of. We discuss each of them:

  1. We included [S. Ryu and Y. Hatsugai, PRB 73, (2006) 245115] as it explicitly links a contribution in the value of the bipartite entanglement entropy to the Zak phase for periodic boundary conditions. As argued in our Sec. 2.3, the disconnected entanglement entropy is not a good topological detector then.
  2. We included both [L. Campos Venuti et al. PRL 96, (2006): 247206] and [L. Campos Venuti et al. PRA 76, (2007) 052328] that point out that edge states can lead to long-range entanglement in low dimensional systems. The two articles are, however, restricted to spin models (bosonic systems) and use the end-to-end concurrence which cannot probe topology (at least, in the form discussed in these articles).
  3. We included [J. Sirker et al., J. Stat. Mech., (2014) P10032] for multiple reasons (similar analytical approach; use an entanglement-based topological criterion; scaling analysis of the entanglement entropy). The latter scaling analysis is in terms of the length of the subset for an infinite or semi-infinite chain. In contrast, our scaling analysis is in terms of the length of the total chain.
  4. We included [N. Sedlmayr et al., Phys. Rev. B 97, (2018) 064304] as it studies the time evolution of the bipartite entanglement entropy after an instant quench.

The referee writes: 2. In the abstract it is written that "To corroborate the topological origin of the quantized values of SD, we show how the latter remain quantized after applying unitary time evolution in the form of a quantum quench, a characteristic feature of topological invariants." I would question that this is a characteristic feature of a topological invariant. For example, if I take the usual invariant for a 1D BDI model, the Zak-Berry phase, it is a property of the ground state. Following a quench the system is very far from its ground state and there is no meaning to use this invariant. It seems to me that this has been mixed up with the definition fo a topological phase which states that all ground states joined by (symmetry preserving) adiabatic unitary evolution (without closing the gap), are topologically equivalent. However it is not time evolution that is referred to here. (See for example Ref [9].) It is also not clear to me that this is a desirable property of a topological invariant. Following a quench the system is far from equilibrium, why should one believe that the topological properties of its ground state are preserved? Nonetheless I agree that this is an interesting observation for SD, I would appreciate more discussion of this point. (See also further related points below.)

Our response: 2. This question is at the crux of our Sec. 5. For emphasis, we added a subsection dedicated to the related explanation in the text (now Sec. 5.1). We have made our statement more specific (indeed, while a valid criterion here, it cannot be applied in general, as the Referee points out - see below). The statement now reads: "...a characteristic feature of topological invariants [associated with the particle-hole symmetry]." This explanation follows [McGinley, M. & Cooper, N. R., Topology of One-Dimensional Quantum Systems Out of Equilibrium., Phys. Rev. Lett. 121, 090401 (2018)] (now cited).

After a local and unitary quench, the time evolved topological ground state of the initial Hamiltonian remains particle-hole symmetric. Thus, any topological invariant associated with the symmetry and extracted from the ground state remains invariant during the dynamic. These non-local invariants include the half-integer Zak phase (over $2\pi$) modulo 1, and, we argue, $S^D$. Unlike the Zak phase modulo $2\pi$ that is the same for both the topological and trivial phase of the SSH model, $S^D$ keeps track of the existence of edge states in the initial state and thus differentiate the two phases despite the dynamics. This last point could not be predicted from Ref.[28] as the Zak phase modulo $2\pi$ of the two phases is $\pi$ and $0$ respectively.

Agreeing on this point, there are two objections the Referee might be concerned about:

-We show that $S^D$ is conserved after diverse independent instant quenches (inducing each an example of local unitary evolution), but does that mean that it will be conserved for all local unitary evolution, i.e. is relevant for some ``real world'' applications?

We argue that it does, as any local unitary evolution can be understood as a succession of independent (and symmetry-preserving) instant quenches, none modifying $S^D$.

-There could be non-local quantities that are not topological invariants and yet are conserved after any local unitary quench, and $S^D$ would be one of them.

This possibility explains why we cannot (and do not) conclude unequivocally that $S^D$ is a topological invariant from this study alone. It remains that this feature of invariance to quenches can be considered as a characteristic of topological invariants, although not exclusive. We strongly believe, however, that the additional invariance of $S^D$ to disorder (presented in Sec. 6) reinforces the topological invariant hypothesis.

Finally, as noticed by the Referee, this property of time-invariance of topological invariants and states with PHS to quenches is precisely why it is difficult to prepare the system in a topological state through unitary evolution. Other means are necessary instead such as breaking the protecting symmetry (e.g. [S. Barbarino et al, Phys. Rev. Lett. 124, 010401 (2020)]) or using dissipation (e.g. [J. C. Budich et al. Phys. Rev. A 91, 042117 (2015)]).

The referee writes: 3. In the first paragraph on p2 the authors write that the TEE "works both in and out of equilibrium". However as afar as I can see tracing the references back, in Hamma et al. [...] which is not referenced directly, they only deal with adiabatic evolution, not quenches or generic non-equilibrium scenarios. In Ref. [6] they are not dealing with the TEE but the usual EE as far as I can see, they they do consider quenches. Could the authors provide a clear reference for this statement?

Our response: 3. We added the missing references for this statement. The TEE is used out-of-equilibrium for fast quenches and the toric code in [Tsomokos, D. I., Hamma, A., Zhang, W., Haas, S. & Fazio, R. Topological order following a quantum quench. Phys. Rev. A 80, 060302 (2009)]. The Rényi-2 version is used in the same context in [Halász, G. B. & Hamma, A. Topological Rényi Entropy after a Quantum Quench. Phys. Rev. Lett. 110, 170605 (2013)] and [Halász, G. B. & Hamma, A. Probing topological order with Rényi entropy., Phys. Rev. A 86, 062330 (2012)].

The referee writes: 4. In the footnote on p2 the authors write that the entanglement spectrum for the Ising and Kitaev model are the same, although only one of them is topological. Do the authors know a counterexample which is not mappable to a topological model?

Our response: 4. We are not sure if we understood what the Referee meant here. Two different patterns of spontaneous symmetry breaking lead to different states that are both non-topological. Yet, they may have the same entanglement spectrum.

The referee writes: 5. At the end of p3 it should say v/w/gg1 for the condition when sites within a unit cell dimerized. For v/w close to 1 coupling both within and between unit cells is similar.

Our response: 5. We now clarify this point in the text.

The referee writes: 6. On p4 the authors write that the chiral S operator is anti-unitary. In fact it is T and C which are anti-unitary, not S.

Our response: 6. According to Ref.[35] that we explicitly call for these definitions, T and S are anti-unitary, while C is not. Thus, what we wrote in the text is right. We nonetheless removed mention of anti-unitarity for chirality, as we do not use the property in any derivation.

The referee writes: 7. Note that relevant to section 3, an semi analytical solution exists for the SSH model, which allows some analytical calculations to be performed. See Byeong Chun Shin, [...].

Our response: 7. We thank the Referee for this suggestion (also contained indirectly in the fourth reference of point 1). It is a valid alternative to our approach in Sec. 3 and likely gives a better fitting for Fig.2 b). As Sec. 3 gives the correct phenomenology and the correct results for the localization length, we believe that the suggested more involved exact solution is not as useful as the one presented in Sec.3 in the context of the article. We nonetheless now mention the reference for the interested reader.

The referee writes: 8. I would appreciate an equivalent of figure 4 for the opposite quench case, to demonstrate the difference. Also, the inset of figure 4a is not explained in the caption.

Our response: 8. We added the requested figures. We added the description of the inset of Fig. 4a.

---

## Round 2 · Referee Report · Anonymous (Referee 2) · 2020-11-12

Report

I thank the authors for the very clear and thorough reply. I think that that all of my questions about the content have been adequately addressed, and the current version also makes clearer what the novelty and achievements are of this particular work. The only question left is whether this is sufficient to satisfy the publication criteria of SciPost Physics Core, and I am happy to say that the authors have also convinced me of this with the clearer presentation of their results (as summarised in their response). I recommend publication.

---

## Round 2 · Author Response

Dear Editor-in-Charge,

We thank you for both your reply and your consideration of our manuscript.
As recommended, we revised the text of the manuscript. We thank both Referees for their comments. Please find our reply to the Referees’
reports and the list of subsequent changes from the previous version of the manuscript to the resubmitted version. Please also find the
resubmitted version of our manuscript.
We hope both you and the referees will find the new version satisfactory.

Yours sincerely,
The authors.

---

## Round 2 · List of Changes

— To answer the main concerns of the two Referees, we significantly rewrote
the introduction and the discussion part of the conclusion, and
added two subsections (5.1 and 7.1). We list more specific changes
below.
— We modified the end of the abstract.
— We modified (added) the titles for section 3, 4, 4.2, 5, 5.1, 5.2, 6, 7.1
— We stress more our main results in the introduction and stressed the
differences with Ref.31 (ex Ref.28) (the first main point of both Referees)
and added Ref.32 and 39 to do so.
— We added Ref. 9-11 (point 3 of the second Referee), 36-37 (point 1.2
of the second Referee).
— We clarified the second paragraph of 2.1 (point 5 of the second Referee).
— We removed a word in the fourth paragraph of 2.1 (point 6 of the
second Referee).
— We corrected a typo in the same paragraph and modified the sentence
accordingly. One reference (ex. Ref 47) was replaced in the process by
Ref.44.
— We added footnote 4 with Ref. 48 (point 1.3 of the second Referee, also
cited later).
— We slightly rephrased the first sentence of 2.3.
— We added two sentences at the end of 2.3 (point 1.1 of the second
Referee).
— We slightly rephrased the first paragraph of 2.4 and corrected a typo
in the second.
— We slightly rephrased the first paragraph of 3
— We added a sentence in the second paragraph of 3 with Ref.59 (point
7 of the second Referee).
— We slightly rephrased the third paragraph of 3
— We added sentences to the first paragraph of 4.1 (point 1.3 of the
second Referee).
— We added section 5.1, added Ref. 64-67, and reformulated the start
of Sec.5 accordingly (point 2 of the second Referee, and also partially
answer the first main concern of both Referees and the second main
concern of the second Referee).
— We added Figs. 4c and d) and added the relevant caption in 5.2 (point
8 of the second Referee). We modified 5.2 to take these figures into
account.
— We added a sentence and Ref. 69 to 5.2 (point 1.4 of the second Referee).
— We corrected a typo on the number of realization mentioned in the
caption of Fig.5.
— We added Sec. 7.1 and modified the discussions and conclusions accordingly
(all main concerns of both Referee). We added the relevant
Ref.77 and 78.
— We now mention where to find the code and data used in the text with
Ref.80.

---

## Editorial Decision

published